# Recent Advances and the Potential for Clinical Use of Autofluorescence Detection of Extra-Ophthalmic Tissues

**DOI:** 10.3390/molecules25092095

**Published:** 2020-04-30

**Authors:** Jonas Wizenty, Teresa Schumann, Donna Theil, Martin Stockmann, Johann Pratschke, Frank Tacke, Felix Aigner, Tilo Wuensch

**Affiliations:** 1Department of Hepatology and Gastroenterology, Campus Charité Mitte and Campus Virchow-Klinikum, Charité—Universitätsmedizin Berlin, Augustenburger Platz 1, 13353 Berlin, Germany; schumannteresa2@gmail.com (T.S.); frank.tacke@charite.de (F.T.); 2Department of Surgery, Campus Charité Mitte and Campus Virchow-Klinikum, Charité—Universitätsmedizin Berlin, Augustenburger Platz 1, 13353 Berlin, Germany; donna.theil@posteo.de (D.T.); leberfunktion@charite.de (M.S.); johann.pratschke@charite.de (J.P.); felix.aigner@bbgraz.at (F.A.); tilo.wuensch@charite.de (T.W.); 3Department of General, Visceral and Vascular Surgery, Evangelisches Krankenhaus Paul Gerhardt Stift, Paul-Gerhardt-Str. 42–45, 06886 Lutherstadt Wittenberg, Germany; 4Department of Surgery, Krankenhaus der Barmherzigen Brüder Graz, Marschallgasse 12, 8020 Graz, Austria

**Keywords:** autofluorescence imaging, clinical studies, endogenous fluorophores, imaging, inflammation, systematic review

## Abstract

The autofluorescence (AF) characteristics of endogenous fluorophores allow the label-free assessment and visualization of cells and tissues of the human body. While AF imaging (AFI) is well-established in ophthalmology, its clinical applications are steadily expanding to other disciplines. This review summarizes clinical advances of AF techniques published during the past decade. A systematic search of the MEDLINE database and Cochrane Library databases was performed to identify clinical AF studies in extra-ophthalmic tissues. In total, 1097 articles were identified, of which 113 from internal medicine, surgery, oral medicine, and dermatology were reviewed. While comparable technological standards exist in diabetology and cardiology, in all other disciplines, comparability between studies is limited due to the number of differing AF techniques and non-standardized imaging and data analysis. Clear evidence was found for skin AF as a surrogate for blood glucose homeostasis or cardiovascular risk grading. In thyroid surgery, foremost, less experienced surgeons may benefit from the AF-guided intraoperative separation of parathyroid from thyroid tissue. There is a growing interest in AF techniques in clinical disciplines, and promising advances have been made during the past decade. However, further research and development are mandatory to overcome the existing limitations and to maximize the clinical benefits.

## 1. Introduction

Endogenous fluorophores exist in human tissue and emit light upon excitation by a suitable wavelength, a phenomenon called autofluorescence (AF). AF is detectable in vitro and in vivo in real time without specific exogenous labeling. A range of endogenous fluorophores have been identified, located intra- or extracellularly. Amongst these, molecules like aromatic amino acids, cytokeratins, collagens, elastin, NAD(P)H, flavins, fatty acids, vitamin A, porphyrins, and lipofuscin have been the most extensively studied [1]. Depending on their characteristic excitation and emission spectra and physiological and pathophysiological tissue expression, altered AF signals appear to be potentially useful for diagnostic purposes. AF appearance, distribution, and spectroscopic range are thereby directly dependent on the natural conditions of assessed tissues. Compared to synthetic fluorophores (e.g., antibody conjugates), the emission spectra of endogenous fluorophores are often broader and can interfere with the fluorescence signal of antibody staining. We recently described AF caused by lipofuscin as an intrinsic biomarker for colonic inflammation [2], a finding we came across by chance, when AF distorted the specific immunofluorescence staining [3]. Thus, tissue AF is both advantageous as a powerful intrinsic biomarker and disadvantageous as a confounding signal in antibody-based immunofluorescence staining. Its specific occurrence and sensitivity in reflecting pathophysiological changes renders AF a promising diagnostic tool. AF imaging (AFI) has been evaluated in several clinical fields. A range of imaging systems have been developed, which are now commercially available and readily useable in research and clinical applications. Depending on the individual AFI system, light with a specific excitation wavelength range is used to elicit the fluorescent light emission from the targeted endogenous fluorophore. AFI systems’ diagnostic readouts differ; for example, some devices display AFI data as numeric values classified according to a reference standard. Other systems overlay fluorescence pseudo-colors onto live images or direct illumination and visualization of target tissue. The most extensive studies have been conducted in ophthalmology, where fundus AF as a non-invasive retinal imaging modality is meanwhile well-established for diagnosing and monitoring retinal diseases [4,5].

AFI is a dynamically developing field in biomedical research. This review provides an up-to-date systematic literature overview on studies reporting AF techniques in several clinical disciplines, excluding ophthalmology, during the past decade. Specifically, we herein focus on clinical application scenarios, the diagnostic accuracy of AFI techniques, technical insights into common AF systems, and possible advantages and disadvantages. In addition, a section is dedicated to biomolecules acting as endogenous fluorophores, as well as their AF properties and diagnostic value.

## 2. Results

### 2.1. Autofluorescence Imaging (AFI) in Gastrointestinal Diseases

Endoscopic assessment of the gastrointestinal (GI) tract represents the ideal dark environment for the ambient light-sensitive AFI technique. The AFI mode is a ready-to-use technique, commercially available for standard endoscopes and used to complement conventional white-light endoscopy (WLE). The so-called endoscopic trimodal imaging (ETMI) combines standard WLE with AFI and narrow-band imaging (NBI) to increase the detection rate of hardly visible or invisible lesions within the GI tract. In the AFI mode, fluorescence is detected upon blue light excitation (390–470 nm) (Figure 1) [6]. Intestinal AF signals were found to show peak intensities at 500–550 nm and 570–620 nm, likely because of the presence of several fluorophores [2,7]. AFI labels suspicious tissue regions in pseudo-purple, whereas normal regions appear green. The following paragraphs summarize the published evidence on the diagnostic abilities of AFI in inflammatory and neoplastic tissue derangements of the GI tract.

#### 2.1.1. AFI in Gastroesophageal Reflux Disease

The ability of AFI to detect mucosal inflammation of the esophagus was investigated by Wang et al. in 82 patients undergoing assessment for gastroesophageal reflux disease (GERD) [8]. Compared to standard WLE, AFI showed higher sensitivity and accuracy in detecting mucosal inflammation (sensitivity: 77% vs. 21%, accuracy: 67% vs. 52%) [8]. Whether AFI can distinguish patients with pathologic non-erosive reflux disease (NERD) from those with functional heartburn (FH) was investigated by Luo et al. [9]. Amongst the 127 screened patients with typical reflux symptoms, 84 patients with normal esophageal appearance on WLE were further evaluated using AFI. Of those 84, 68 showed signs of inflammation on AFI. In the NERD group (classified using the proton-pump inhibitor test), 67 patients (91%) with abnormal esophageal AFI signals were detected, while only one patient in the FH group was positive on AFI. The sensitivity and specificity of AFI in differentiating NERD from functional heartburn were 91% and 90%, respectively, leading the authors to conclude that AFI represents a complementary method in evaluating patients with NERD and functional heartburn.

#### 2.1.2. AFI in Intestinal Inflammation

Osada et al. presented an analysis of 286 WLE and AFI images of the same colonic sites in 42 ulcerative colitis (UC) patients. By adding an RGB (red, green, and blue) additive color model in silico, a strong inverse correlation between the green color component of AFI images and the severity of the mucosal inflammation was found [10]. In a similar prospective study by Moriichi et al., 135 WLE and AFI images from the same lesion in 43 patients with UC were evaluated by eleven endoscopists [11]. The AFI intensity significantly inversely correlated with the histologically graded severity of inflammation, and AFI was superior to standard white-light video endoscopy (SVE) regarding detection of inflammation (85% vs. 79%). This study further showed that AFI is particularly useful for detecting colonic inflammation by less experienced endoscopists (<500 standard and <10 AFI procedures). Therefore, AFI appears to be primarily beneficial for the purposes of training less experienced endoscopists, but may not increase diagnostic accuracy in detection of colonic inflammation by expert endoscopists.

#### 2.1.3. AFI in Upper GI Neoplasia

A frequent complication of chronic mucosal inflammation and relapsing tissue regeneration is neoplastic malignancy. In a randomized study on 87 patients with Barrett’s esophagus, Curvers et al. investigated whether ETMI could increase the detection rate of high-grade dysplasia or early carcinoma compared to SVE only [6]. The ETMI procedure was conducted in the following order: First, high-resolution endoscopy (HRE) was applied, followed by AFI. All suspect lesions detected by HRE or AFI were subsequently inspected by NBI. The detection of high-grade dysplasia and early esophageal adenocarcinoma in Barrett’s esophagus was significantly higher in the ETMI group compared to SVE alone, but with a huge false-positive rate (71%). Serial detailed inspection with NBI after AFI was able to reduce the false-positive rate to 48%, which is in the same range as that reported by Giacchino et al., who assessed ETMI performance in detecting high-grade dysplasia and early esophageal adenocarcinoma in 42 patients with Barrett’s esophagus [12]. In a further multicenter randomized trial by Curvers et al., 99 patients with Barrett’s esophagus and a confirmed low-grade intraepithelial neoplasia underwent ETMI and SVE. The results show that, despite ETMI detecting 22 more suspect lesions for targeted biopsies, there was no significant difference in the overall histological yield (targeted + random) between ETMI and SVE [13]. The false-positive grading of dysplastic areas by AFI might depend—at least in part—on the observer’s experience. When AFI experts and non-experts evaluated the same AFI images, recorded during Barrett’s surveillance endoscopies, experts identified dysplasia more accurately than non-experts. Accuracy improved with the combined assessment of AFI and HRE images, independent of the observer’s experience level. AFI was also evaluated in combination with NBI for its ability to detect gastric metaplasia and mucosal atrophy in patients with dyspepsia [14]. The combined endoscopic assessment with AFI and NBI identified more intestinal metaplasias (68% vs. 34%) and mucosal atrophies (32% vs. 13%) than SVE alone, although the specificity of AFI was low (23% for intestinal metaplasia). Lim et al. assessed the accuracy of AFI in detecting gastric intestinal metaplasia at 125 sites from 20 patients with a history of gastric metaplasia [15]. The highest accuracy was found for the probe-based confocal laser endoscopy (88%), followed by AFI (69%) and SVE (65%). AFI was also evaluated for detecting new squamous mucosal high-grade neoplasias in patients with a history of esophageal neoplasia or head and neck cancer. In a study by Ishihara et al., AFI was compared to chromoendoscopy as the reference standard method in 364 patients undergoing endoscopic examination, either by an experienced (*n* = 180) or by a less experienced endoscopist (*n* = 184) [16]. The sensitivity of AFI in detecting neoplasia was 71% when assessed by an experienced endoscopist and 50% when assessed by a less experienced endoscopist. Furthermore, AFI showed a significantly higher sensitivity in detecting lesions >10 mm in diameter than smaller ones.

As the vast majority of studies reveal, AFI appears not to offer a benefit in detecting mucosal neoplasia compared to standard endoscopic and histological examinations. However, it might be valuable in combination with NBI or HRE to support the interpretation of suspect areas, specifically for less experienced observers.

#### 2.1.4. The Accuracy of AFI for the Diagnosis of Colorectal Cancer

While AFI endoscopy is a reliable method to detect colonic inflammation, the studies investigating its sensitivity for the endoscopic differentiation of dysplastic lesions yielded controversial results. McGinty et al. found that an optimal contrast between healthy and cancerous colon tissue is found with excitation at 355 nm [17].

AFI was applied in several studies to detect colonic dysplasia in patients undergoing surveillance colonoscopies. Van den Broek et al. examined 50 UC patients for neoplastic lesions, either with AFI followed by WLE (*n* = 25) or vice versa (*n* = 25) [18]. All neoplastic lesions could be identified by AFI, but only 50% by WLE (*p* = 0.036); no further lesions were detected by random biopsies. The performance of serial assessment was also studied by Rotondano et al., who tested the ability of ETMI to detect colorectal neoplasia in a prospective randomized trial on 94 patients scheduled for surveillance colonoscopy with a personal history of neoplasia or a positive family history of colorectal cancer [19]. All colonic segments were examined by HRE and AFI, and detected lesions were removed and histologically analyzed. The accuracy of AFI in detecting colonic adenomas was lower than for NBI (63% vs. 80%, *p* < 0.001), but the combined use of AFI and NBI yielded an improved accuracy for adenoma detection (84%) compared to HRE. When Vleugels et al. assessed 210 patients with long-standing UC, the sensitivity for real-time prediction of dysplasia was 76.9% when ETMI was applied [20]. In a separate analysis of diagnostic test accuracies, AFI had the highest sensitivity (91.7%) and highest negative predictive value (98.7%) compared to WLE, NBI, and chromoendoscopy [20]. A further data analysis of the above-mentioned multicenter trial found no differences in “per-biopsy yield” of dysplasia detection when AFI and chromoendoscopy with either 0.1% methylene blue or 0.2% indigocarmine solution were compared [21]. In contrast, Kuiper et al. found no improved adenoma detection rates when using ETMI compared to SVE in 234 patients scheduled for surveillance colonoscopy because of a history of adenomatous polyps, a history of surgically treated colorectal cancer, or with a positive family history of colorectal cancer [22].

The performance of AFI in differentiating between colorectal hyperplasia, neoplasms, and the grade of dysplasia was also examined by several studies. Moriichi et al. analyzed AFI images of 158 lesions from 67 patients by calculating the F index (ratio of the green fluorescence divided by the red reflection) as a quantitative value of the AF images, and found AFI to be valid for predicting the dysplastic grade, regardless of tumor shape [23]. In contrast, the visual classification of the AFI signal was significantly associated with the size of the lesion, which might lead to incorrect classifications based on visual examination. In a prospective, multicenter, randomized controlled trial by the group of Takeuchi et al., the efficacy of AFI or WLE in detecting and discriminating between flat (nonpolypoid) and polypoid colorectal lesions was evaluated in 802 patients undergoing colonoscopy [24]. The detection of flat neoplasms was significantly higher with AFI compared to WLE, while detection rates of diminutive (1–5 mm) polypoid neoplasms were higher with WLE. Experienced colonoscopists (>5000 colonoscopies) detected flat neoplasms more frequently in the AFI group than in the WLI group; however, less experienced colonoscopists (<5000 colonoscopies) did not. Matsumoto et al. assessed AF intensities of apparently flat, coarse granular mucosa and visible protruding lesions in 48 UC patients using first WLE, with suspect regions subsequently evaluated by AFI [25]. The positive rate of dysplasia in protruding lesions was significantly higher in low AF than in high AF, while the rate of dysplasia in flat lesions did not differ between low AF and high AF. AFI is reliable in detecting dysplasia in protruding regions; however, especially in flat lesions, simultaneous assessment by chromoendoscopy or NBI would be indicated. Improved diagnostic accuracy in discriminating colon adenoma from hyperplastic polyps was observed by Sato et al., when conventional HRE images for 183 patients were additionally reviewed with AFI and NBI (75.9% vs. 91.9%) [26]. These improvements appeared independently of the observers’ experience, while the accuracy in detecting lesions ≤1 cm in size was improved in the group of less experienced endoscopists (performed <500 colonoscopies). In the study by Ignjatovic et al., experienced colonoscopists with (“expert” group, *n* = 4) or without (“nonexpert” group, *n* = 5) knowledge of advanced imaging (ETMI) were asked to evaluate 320 endoscopic images and state a clinical decision on the likely histology of polyps (adenoma vs. hyperplastic polyp). The “nonexpert” group received training on advanced imaging techniques, including 20 images of polyps from all four included imaging modalities (AFI, NBI with and without magnification, and white-light colonoscopy). AFI accuracy for prediction of polyp histology was 53% for “expert” and 32% for “nonexpert” colonoscopists. The specificity and accuracy of AFI were lower compared to WLE for both “experts” and “nonexperts”, while sensitivity and overall accuracy were higher with NBI compared to WLE. These results indicate that AFI alone might not be sufficient to safely detect adenomas or hyperplastic polyps in the colon. However, for experienced colonoscopists, training in advanced imaging techniques appears to be fast and effective.

In addition to equipping commercial endoscopes, specific technical setups might improve the performance of AFI. Takeuchi et al. tested a transparent hood on the tip of the colonoscope to detect lesions behind the colonic folds by turning the folds over. This randomized controlled trial was carried out on 561 patients who underwent colonoscopy for verification of a positive fecal occult blood test or surveillance colonoscopy after resection of colorectal neoplasms [27]. The neoplasm detection rate was highest when AFI with the transparent hood was applied, compared to WLE or AFI without the transparent hood. Kuiper et al. tested an AF spectroscopy device incorporated into standard biopsy forceps. Colonoscopies from 87 patients with 207 small colorectal lesions were evaluated, and the accuracy of this AFI technique alone or of the algorithm combining AFI with HRE proved to be insufficient for in vivo differentiation between adenomatous and non-adenomatous colorectal lesions [28].

In conclusion, AFI as part of the ETMI procedures is beneficial in improving diagnostic accuracy in detecting colonic neoplasia compared to ordinary WLE. However, with its current technical abilities, ETMI cannot replace random biopsies for histological verification and characterization of neoplastic lesions. Multiple influences, such as technical setup or the observer’s experience, affect the AFI outcome.

### 2.2. AF Bronchoscopy in Pulmonary Diseases

Similar to AFI of the GI tract, AF bronchoscopy (AFB) is an endoscopic push-button technology for improved visualization of neoplastic alterations of the bronchial mucosa. The light source emits excitation light in the 390–440 nm range, and the emitted fluorescence is processed to a pseudo-color image showing malignant mucosa in magenta and normal mucosa in green. A randomized exploratory study compared AFB with white-light bronchoscopy (WLB) to detect pulmonal epithelial changes like vascular abnormalities, suspicious preinvasive lesions, and tumors in 29 patients scheduled for a therapeutic or diagnostic bronchoscopy [29]. Vascular abnormalities were detected most frequently by high-definition bronchoscopy compared to AFB (1.33 ± 0.29 vs. 0.12 ± 0.05 suspicious sites per patient, *p* = 0.003). Suspicious sites of preinvasive lesions were most frequently detected by AFB (0.74 ± 0.12 sites per patient), compared to 0.17 ± 0.06 for both WLB and high definition bronchoscopy (*p* = 0.003). Tumors were detected equally by all modalities. In a prospective, non-randomized trial on 104 lung cancer patients, Zaric et al. assessed the sensitivity and specificity of AFB in detecting the tumor extent and margins [30]. In 14.4% of patients, AFB revealed a greater extent of the tumor than WLB did, which led to adjustment of the therapeutic decision in 11.5% of cases. The sensitivity, specificity, and positive and negative predictive values of AFB in the assessment of tumor extension were 93%, 92%, 92%, and 93%, all higher than the corresponding results for WLB (84%, 79%, 77%, and 85%). In a study assessing patients with head, neck, or bronchus tumors (*n* = 66) using WLE and AFB, AF analysis led to changes in clinical management in four patients compared to examination only by WLE [31]. Tremblay et al. investigated the prevalence of lung tissue abnormalities detected by AFB following lung cancer screening with low-dose computed tomography (LDCT) in a cohort of 1300 patients at high risk for lung cancer [32]. In addition to the 56 lung cancers detected by the LDCT scan, two cases of CT occult malignancies were detected by AFB. The authors concluded that the prevalence of CT scan occult malignancy detected by AFB is too low and does not justify the incorporation of AFB into a lung cancer screening program. The recently published LungSEARCH trial reported results of a five-year surveillance follow-up of patients with a chronic obstructive pulmonary disease (*n* = 1568) who underwent AFB in addition to LDCT screening. The patients who were selected based on an abnormal annual sputum cytology had no benefit in terms of early malignancy detection by the tested lung cancer screening algorithm [33]. In contrast, other studies showed the potential of AFB to detect ischemic damage and to be a useful diagnostic tool for the early detection of acute cellular rejection after lung transplantation [34,35]. Furthermore, AFB is a valid technique for directed biopsy sampling in smokers with bronchial dysplasia [36].

These controversial findings and conclusions warrant further studies to identify the full diagnostic potential of AF techniques in pulmonary disease and to pinpoint the clinical scenarios where AFI can make the difference and improve patients’ treatment.

### 2.3. AF for Non-Invasive Cardiovascular Risk Prediction

#### 2.3.1. AF of Advanced Glycation End Products

Advanced glycation end products (AGEs) are glycated proteins, lipids, or nucleic acids with disrupted physiological functions and that promote proinflammatory signaling via the AGE receptors (RAGE) [37]. The non-enzymatic formation of AGEs occurs in normal metabolism and aging, but is accelerated in metabolic and degenerative diseases. One group of AGEs is that of collagen adducts, which were found in the 1980s to be fluorophores, showing AF characteristics (excitation maxima at 370 nm) and detectable in skin tissues [38]. The technical development of a skin AGE AF detection device led to a range of studies testing the prognostic ability of noninvasive skin AF (SAF) measurements in clinical fields, such as diabetes, kidney disease, or pulmonary disease (Figure 2). The commercially available AGE Reader (DiagnOptics) illuminates approximately 4 cm^2^ of skin on the forearm, with SAF calculated as the ratio between the total emission intensity (420–600 nm) and the total excitation intensity (300–420 nm). The measured value is displayed in arbitrary units (AU). In a validation study of SAF against skin biopsy AF, Meerwaldt et al. found correlations between SAF and collagen-linked, pentosidine, and *N*ε-(carboxyethyl)lysine AF for the skin biopsies [39]. In 52 patients with diagnosed coronary heart disease undergoing a coronary bypass operation, Hofmann et al. assessed SAF and analyzed AGE AF in isolated collagen fractions of vein material [40]. SAF and vessel stiffness correlated significantly with the AGE AF of the collagenase-digestible collagen fraction of graft vein material. Furthermore, in diabetic patients, SAF was correlated with age, diabetes duration, HbA1c, and plasma creatinine levels [39], and higher SAF was also found to be associated with decreased muscle strength in older adults [41]. These results proved SAF to be a reliable predictor of vascular AGE modifications and accumulation.

#### 2.3.2. SAF Determination in Diabetes

In a prospective cohort study of 563 patients with a median diabetes duration of 13 years and a follow-up of 5.1 years, SAF was associated with macrovascular events, whereas HbA1c was associated with microvascular events [42]. Liu et al. demonstrated an association between SAF and diabetic complications like retinopathy, nephropathy, coronary heart disease, cerebrovascular disease, and peripheral artery disease in diabetic patients (*n* = 118) [43]. However, in their study, HbA1c did not predict the occurrence of vascular complications. In a five-year follow-up study of obese patients after bariatric surgery, weight loss and metabolic improvements were not accompanied by changes in SAF, which remained higher compared to the non-obese control group [44]. A consequence of AGE accumulation is arterial stiffening, which may explain the increased cardiovascular risk in diabetes. Ninomiya et al. revealed a significant association between SAF and flow-mediated vasodilation (r = −0.259, *p* = 0.002), a subclinical atherosclerosis marker, in 140 Japanese patients with type 1 or type 2 diabetes [45]. The Maastricht study included 862 participants and reported an association between SAF and carotid–femoral pulse-wave velocity and pulse pressure, both measures of arterial stiffness, which was more pronounced in the group of patients with type 2 diabetes [46]. The hypothesized mechanisms behind this relationship are AGE-formed cross-links between collagen in the arterial wall or myocardium, quenching of nitric oxide with increased smooth muscle cell tone, or proinflammatory reactions via RAGE. The relationship between AGE-modified collagen in cardiac tissue and SAF was described by Hofmann et al. in 72 patients with coronary artery disease [47]. Willemsen et al. reported an association between SAF and the aerobic capacity measured using peak oxygen uptake in diabetic (*n* = 49) and non-diabetic (*n* = 156) patients with diastolic heart failure [48]. Diabetic patients had elevated SAF values and a significantly lower exercise capacity than non-diabetic heart failure patients. This workgroup further revealed that in patients with diastolic dysfunction, an extended anti-hypertensive treatment with the angiotensin II type 1 receptor antagonist eprosartan significantly reduced blood pressure, but did not alter AGE levels [49]. However, patients with SAF values below the median showed a larger improvement in diastolic function, assessed by echocardiography in response to any anti-hypertensive treatment, compared to patients with higher SAF. A seven-year follow-up study demonstrated that SAF predicts cardiovascular events in patients (*n* = 232) with type 1 diabetes [50]. It was also shown in 476 type 1 diabetic patients with a median disease duration of 24 years that insulin resistance was positively associated with SAF [51]. A correlation between SAF and mean HbA1c was even found in children with type 1 diabetes [52]. Given the obvious evidence of the relationship between blood glucose homeostasis and SAF values, measuring SAF could be applicable for risk prediction of developing diabetes. This hypothesis was tested by van Waateringe et al. in a large prospective trial on the general population (*n* = 72,880) [53]. After a four-year follow-up, 1056 participants (1.4%) developed type 2 diabetes. These subjects displayed significantly higher SAF (mean 2.13 ± 0.45 AU) compared to non-diabetic subjects (mean 1.90 ± 0.42 AU). A comparable number of subjects with elevated SAF (2.18 ± 0.47 AU) had a cardiovascular event (*n* = 1258, 1.7%) or died (*n* = 928, 1.3%, SAF 2.33 ± 0.52 AU). Subjects without an incident had a mean SAF value of 1.90 ± 0.42 AU. Smit et al. tested an SAF-based decision tree in detecting diabetic patients in 218 persons with one or more metabolic syndrome criteria [54]. The SAF-based decision tree was superior to fasting plasma glucose and non-inferior to HbA1c in detecting diabetes and impaired glucose tolerance in intermediate-risk persons. Independent of their diabetic status, AGE accumulation was found in a prospective observational study (*n* = 764) to be associated with cognitive performance, showing delayed word recall and response inhibition in individuals with higher SAF values [55]. Finally, Bakker et al. found no differing SAF values in patients with type 1 diabetes with or without concomitant coeliac disease [56].

A reduction of AGE is desirable and could be achieved by pharmacological treatments to break AGE crosslinks. The effects of a 36 week treatment with the AGE-breaker alagebrium was investigated by Hartog et al. in a prospective, randomized, double-blind, placebo-controlled study in 102 patients with chronic heart failure [57]. Alagebrium failed to improve cardiac function, exercise capacity, or SAF. Another potential drug supposed to block hyperglycemia-induced AGE formation, benfotiamine, was evaluated by Stirban et al. in 22 patients with type 2 diabetes [58]. The tested six-week treatment did not influence SAF values [58]. A cross-talk between muscle and skin AGE accumulation was suggested by Li et al., who found a significant inverse relationship between serum levels of the myokine irisin and SAF values in 362 Chinese type 2 diabetic patients [59]. The influence of a nutritional intervention with a whey protein isolate on SAF in overweight subjects with impaired blood glucose levels was also assessed. Supplementation with 40 g/day whey protein isolate for 12 weeks did not affect SAF, inflammation, or blood glucose markers [60].

AGE accumulation in the skin was suggested to influence the synthesis of vitamin D, which is solely produced in the skin. A large prospective trial (*n* = 2746) found an inverse association of serum 25-hydroxyvitamin D with SAF in diabetics and non-diabetics [61]. In another study of 245 patients with type 2 diabetes, SAF values were significantly higher in patients with low serum 25-hydroxyvitamin D levels (<50 nmol/L) compared to patients with serum 25-hydroxyvitamin D > 75 nmol/L [62]. In the same study, a group of 107 patients received a 25-hydroxyvitamin D supplement for 6 months, resulting in a significant increase in 25-hydroxyvitamin D plasma levels. However, 25-hydroxyvitamin D supplementation had no effect on SAF in these patients. No correlation between 25-hydroxyvitamin D plasma levels and the accumulation of AGEs was also reported by Stürmer et al. in non-diabetic healthy persons or hypertensive patients [63].

#### 2.3.3. SAF Determination in Chronic Kidney Disease

AGEs are uremic toxins that occur in chronic kidney disease (CKD) due to impaired AGE clearance and increased formation caused by the unbalanced redox homeostasis with elevated levels of reactive oxygen species and free radicals. In CKD, elevated SAF (>2.31 AU) is an independent predictor of disease progression in pre-dialysis patients [64]. Thus, SAF might be used as a clinical biomarker for risk stratification of CKD patients. The highest SAF values exist in patients on dialysis. In over 300 hemodialysis patients (mean age 65.7 ± 15.1 years) with a mean dialysis vintage of 65.1 months (range 1–413), Nongnuch et al. found a mean SAF of 3.27 ± 0.96 AU, and patients with SAF above the mean had a higher risk of death during the 30 month follow-up [65]. In a separate analysis of the same study cohort, SAF was significantly lower in vegetarians than in non-vegetarians (2.71 vs. 3.31 AU, *p* = 0.002), which led the authors to hypothesize that the lower dietary intake of preformed AGEs by vegetarians might cause this difference [66]. In patients on peritoneal dialysis, Macsai et al. found SAF values over 3.61 AU as a predictor of mortality during the 36 month follow-up of 198 Caucasian patients [67]. SAF in peritoneal dialysis was also associated with cardiovascular morbidity in over 2000 patients [68]. Crowley et al. showed significantly lower SAF values in kidney transplant recipients (2.81 ± 0.64 AU), compared to patients on hemodialysis (3.73 ± 0.88 AU) or peritoneal dialysis (3.57 ± 0.75 AU), respectively [69]. Whether kidney function can be predicted by SAF was investigated by Schutte et al. in 471 patients with peripheral artery disease and preserved kidney function [70]. An increase in SAF during the three-year follow-up period was not associated with kidney function decline (assessed by the estimated glomerular filtration rate). AGEs were also found to be linked to the pathophysiology of bone mineral disorder in CKD. Franca et al. reported a negative correlation (r = −0.497; *p* = 0.026) of SAF with serum parathormone, the main bone turnover marker, in 20 hemodialysis patients undergoing dialysis for at least three months [71].

#### 2.3.4. SAF Determination in Chronic Obstructive Pulmonary Disease

SAF was found to be higher in patients with chronic obstructive pulmonary disease (COPD) and a history of cigarette smoking (*n* = 202), compared to 83 elderly (age between 40–75 years) and 110 young (age between 18–40 years) healthy smokers and never-smokers [72]. SAF values differed significantly between all three groups (2.5, 1.8, 1.2 AU, *p* < 0.001), and higher SAF values were associated with higher numbers of pack-years, independently of gender, age, or comorbidities. SAF values did not differ between COPD patients of all severity stages (A−D), classified by the Global initiative for chronic Obstructive Lung Disease (GOLD) stages [73]. A significant age-dependency of SAF values was reported by Kubo et al. when two groups of 145 younger and 56 elderly subjects (age ≥65 years) with no history of pulmonary disease were compared. In addition, SAF and smoking history (pack-years) were significantly independent factors associated with lung function (forced expiratory volume in one second/forced vital capacity ratio) in the elderly group [74]. Individual AGE formation and accumulation in the skin may be influenced by genetic factors. Hoonhorst et al. identified the single nucleotide polymorphism rs915895 (located 38188 bp upstream of the AGER gene) as associated with AGE accumulation in the skin [75]. These findings add an additional layer of complexity in understanding the causality of AGE formation, SAF values, and the pathogenesis of pulmonary diseases.

### 2.4. AF in Oral Medicine

In oral medicine and dentistry, AF-based devices are commercially available as adjunctive tools to detect (pre)malignant lesions, dentin characteristics, or oral bacteria abundance. Kanchwala et al. characterized the AF spectra of the oral mucosa, using a spectroscopic system consisting of a nitrogen laser and a fiber optic probe, in 52 patients with squamous cell carcinoma or premalignant lesions and different cycles of post-operative radiotherapy [76]. The AF spectra of healthy squamous tissues showed an intense signal at the 460 nm band, while the intensity of the 635 nm band was highest in the spectra from oral squamous cell carcinoma lesions. Furthermore, in patients receiving radiation therapy, the AF intensity of the 460 nm band increased with the number of radiation cycles, as also occurred after surgical treatment. Several studies used the VELscope (Visually Enhanced Lesion scope, LED Dental Inc., Burnaby, BC, Canada) as an oral AFI system to differentiate between benign, dysplastic, and malignant oral mucosa lesions, which are not visible using WL. This device uses a metal-halide arc lamp plus a system of filters and reflectors, producing blue light between 400 and 460 nm. Emission is detected at >500 nm. The loss and shift of tissue AF are indicators for malignant or dysplastic tissue abnormalities. An accordance of 78.9% between the AFI result and the histopathology of suspect tissues areas was found by Sawan et al. in a cohort of 748 patients, while 21.1% of pathological malignant lesions were only grouped as benign by the VELscope [77]. Hanken et al. examined the diagnostic accuracy of VELscope in 120 patients with suspicious oral premalignant lesions and found an increased sensitivity (+22.0%) and specificity (+8.4%) compared to WL only [78]. Higher sensitivity (but lower specificity) in detecting oral cancer lesions in a cohort of 123 patients with premalignant oral lesions was reported by Rana et al. [79]. By including AFI in a population screening for oral cancer and oral potentially malignant disorders, Simonato et al. found that AFI had a higher sensitivity (94.4%) and specificity (96.2%) over conventional oral examination (83.3% sensitivity and 95.1% specificity), and should therefore be considered as an adjunctive method for early diagnosis of high-risk oral lesions [80]. These positive findings, indicating additional diagnostic accuracy, could not be reproduced by Koch et al. when 78 patients with suspicious oral mucosal lesions were assessed for oral dysplasia and carcinoma using the VELscope [81]. A study by Mehrotra et al. compared the performance of a commercial AFI to a chemiluminescent device (VELscope vs. ViziLite Plus) in detecting premalignant oral lesions [82]. Clinically innocuous lesions were detected by conventional overhead light and subsequently assessed by one of the study devices, biopsied, and histologically analyzed. None of the methods showed a benefit in identifying precancerous and cancerous oral lesions; however, the study design lacks proper control conditions for an unbiased assessment of AFI’s diagnostic potential.

In a small retrospective case series, Cherry et al. assessed the ability of AFI to detect potentially malignant oral disorders in 15 patients with existing high-risk lesions, most commonly leukoplakia, undergoing regular surveillance examinations [83]. In contrast to the sole subjective VELscope imaging, the authors of the study quantified the red-to-green pixel intensity of the AF images. This analysis revealed a positive result in eight of nine cases that required surgical intervention, indicating that the use of AFI in surveillance monitoring of patients with potentially malignant oral disorders may assist the clinician in decision-making. Another commercially available medical device for AF detection, the GOCCLES^®^ (Pierrel Pharma Srl, Capua, Italy) eyeglasses, are a pair of glasses equipped with filters that highlight AF when the oral mucosa is illuminated by dental curing light (400–500 nm). Moro et al. used GOCCLES^®^, following a naked eye inspection, to detect lesions in patients at risk for oral cancer [84]. Suspect areas were biopsied and histopathologically evaluated. The device identified 31 of 32 true positive lesions, and 56.7% of the lesions showed a greater extension when observed by GOCCLES^®^, supporting the benefits of AF visualization as a meaningful complementary inspection.

AF might also be applicable for the differential diagnosis of pathological changes in the laryngeal epithelium. Winiarski et al. conducted an in vitro study assessing the AF characteristics in histologically evaluated laryngeal tissue specimens [85]. While the emission spectra upon excitation at 290 nm and 370 nm were similar in healthy and cancerous tissues, the fluorescence intensities were significantly lower in cancerous tissue compared to healthy tissue taken from the same patient.

AF techniques are further available for the detection of dental caries, which are caused by metabolic processes between the dental biofilm on the enamel surface, leading to demineralization of hard dental tissue and therefore loss of natural fluorescence [86]. In addition, bacterial plaque produces porphyrin in carious lesions that emit AF at 635 and 680 nm under proper excitation, which is distinguishable from sound dentin, as healthy dentin shows no red emission [87,88]. Dental carious lesions furthermore show discoloration that affects AF signals, caused by products of the Maillard reaction, a chemical reaction between amino groups, reducing carbohydrates and forming brown-colored polymers [87]. Several commercial medical devices are available and were studied for caries diagnosis. Schwass et al. used the DIAGNOdent™ device (KaVo Dental GmbH, Biberach, Germany), which excites tissue at 655 nm and displays the measured AF as a quantitative numeric value (0–99), allowing ablation only of infected dentin [87]. Another novel imaging configuration called near-infrared light transillumination (NILT) provides excitation light at 780 nm and may be more suitable for proximal early caries lesions [89]. Using the NILT properties with the DIAGNOcam™ device (KaVo Dental GmbH) was indeed superior compared to the DIAGNOdent™ device (excitation at 655 nm) in detecting occlusal caries [90].

AF has also been tested to differentiate between sound, active, and arrested carious dentin. Terrer et al. used an intraoral LED camera that emits excitation light at 455 nm, and found that AF properties enabled differentiation between sound and carious dentin, as active caries showed higher AF brightness values than arrested caries [91]. Banjeree et al. used confocal fiber-optic micro-endoscopy (CFOME), an in-situ tool (excitation wavelength 488 nm and emission detected >500 nm), to compare the efficacy of caries removal techniques. Their results collectively show that microhardness of carious dentine can be quantified by AFI and, therefore, CFOME might have promising clinical potential for evaluating dentin characteristics [92,93].

### 2.5. AF of the Thyroid and Parathyroid Glands

The identification of parathyroid glands (PTGs) in thyroid surgery is crucial to avoid postoperative hypocalcemia as a consequence of the missing parathyroid hormone. In contrast, accurate localization of PTGs during parathyroid surgery for primary hyperparathyroidism is crucial to avoid persistent hypercalcemia [94]. When excited in the near-infrared (NIR) wavelength spectrum, PTG tissues can be detected by the emitted light of a not-yet-identified fluorophore. Paras et al. first described the application of the concept to localize PTGs non-intrusively and intra-operatively in real-time in 2011 in a pilot study on 21 patients undergoing (para-)thyroid surgery [95]. In all cases, excitation at 785 nm led to a detectable AF peak at 820–830 nm. Since then, the AF properties of PTGs have been confirmed by other explorative studies reporting significantly higher AF emission intensities from the PTG than the thyroid gland and surrounding tissues [96,97].

Initial studies used spectroscopy for AF detection, which required direct contact with the relevant tissue [95,98,99]. More recent use of NIR-imaging camera systems has the advantage of enabling contactless working and provides a large surgical field of view. Ladurner et al. modified a commercially available NIR imaging system (Karl Storz SE & Co. KG, Tuttlingen, Germany), which was originally designed for indocyanine green (ICG) imaging, applicable in laparoscopic surgery and showing PTG detection rates of 81.0%–93.3% [100,101,102]. A similar setup was successfully used in a study conducting video-assisted neck surgery (*n* = 5) with combined use of NIR imaging for PTG detection and ICG imaging to prove vascular supply [103]. Kim et al. constructed an experimental setup based on a digital single-lens reflex camera with the aim of designing a PTG detection tool for thyroidectomies. They reached a PTG detection rate of 100% (*n* = 8) and 98.6% (*n* = 38) prior to surgical dissection with the system taking photographic images [97,104] and 98.1% with an improved autofocusing video-imaging system [105]. Feasibility studies on case series of parathyroidectomies found high (98.8%–100.0%) intraoperative PTG detection rates with different technical setups [98,106]. Correspondingly, for a cohort of 310 patients undergoing parathyroidectomy, total thyroidectomy, or thyroid lobectomy, Kose et al. reported that 98% of the PTGs could be identified correctly by AFI, 23% already before visual identification by the surgeon [107]. In a multicenter analysis (*n* = 210), Kahramangil et al. reported an intraoperative PTG detection rate of 97%–99% with NIR AFI in patients undergoing thyroid or parathyroid surgery [108]. Depending on the surgeon, 37%–67% of PTGs were identified intraoperatively by NIR AFI even without previous dissection and visual recognition [108]. Falco et al. found significantly higher PTG detection rates by NIR compared to standard resection under white light in a group of 74 patients undergoing thyroid and parathyroid surgery [109]. In a comparative study of the dye-based indocyanine green (ICG) fluorescence imaging with NIR AFI, both techniques reached similar detection rates [110]. While AFI requires no exogenous dye application, ICG additionally allows assessment of the vascularity of the gland. An often-cited limitation of PTG detection by NIR AFI is the interference of the AF signal with the ambient and overhead lights of the operating room, requiring operating room lights to be switched off for NRI AFI detection. This causes interruptions to the surgical workflow and prolongs operation time (according to Benmiloud et al., a delay of eight minutes to the median operation time) [111]. Several research groups tried to overcome this disruption to surgical routines by introducing specially designed devices. McWade et al. reported on the Overlay Tissue Imaging System (OTIS), which back-projects the collected AF signal in real time in the surgical field-of-view [112]. Thomas et al. designed the now commercially available PTeye device (AiBiomed Corp., Santa Barbara, CA, USA), a handheld fiber-optic probe that gives auditory feedback when in contact with suspected PTG tissue [113]. When comparing the PTeye and OTIS devices with a modified NIR system, similar PTG detection rates of between 97–100% were observed [114]. Another NIR AFI setup was tested on five patients in a study by Serra et al. [115]. With a light-emitting diode, exposed PTG tissue was excited and simultaneously successfully visualized with a night-vision goggle device in all five patients.

While the advantages of PTG detection by NIR AFI during thyroidectomies are widely acknowledged, its impact on clinical outcomes is ambiguous. In a prospective cohort of 269 consecutive thyroidectomies (*n* = 106 with NIR AFI and *n* = 163 without NIR AFI), DiMarco et al. found no reduction in missed inadvertent parathyroidectomies and no difference in hypocalcemia or late hypoparathyroidism when using NIR AFI compared to standard resection under white light [116]. In a before-and-after controlled study on 513 patients undergoing total thyroidectomy, Benmiloud et al. also did not find any difference in inadvertent PTG resection rates when NIR AFI was applied or not [117]. However, an increased mean number of identified PTGs, reduced parathyroid autotransplantation rates, and a significantly lower postoperative hypocalcemia rate (5.2%) were observed in the group of patients with NIR AFI, compared to 20.9% in the control group without NIR AFI. Significantly reduced postoperative hypocalcemia rates were found by Benmiloud et al. when patients (*n* = 241) undergoing total thyroidectomy were randomized to the NIR AFI group compared to the standard-care group without NRI AFI [111]. Furthermore, the number of patients with four identified PTGs was higher and parathyroid autotransplantation was less frequent in the NIR AFI group. More than half (61.6%) of the PTGs could even be identified by NIR AFI before the surgeon saw them with the naked eye. Dip et al. conducted a randomized controlled trial on patients undergoing total thyroidectomy (*n* = 170) and found a trend towards lower postoperative overall hypocalcemia rates (8.2% vs. 16.5%) when PTGs were intraoperatively detected by NIR AFI (*p* < 0.103), compared to white-light (WL) inspection only [118]. However, severe hypocalcemia, defined as a serum calcium level ≤7.5 mg/dL, was observed in 1.2% in the NIR AFI group vs. 11.8% in the WL group (*p* = 0.005).

AF is unevenly and not ubiquitously present in PTGs, representing a risk of overlooking PTGs during surgery even when NIR AFI is applied. The data on NIR AFI positivity for PTGs are controversial. Various studies reported that both healthy and diseased PTGs emit AF light upon excitation [102,119,120]. DiMarco et al. found that nearly 10% of PTGs do not show AF in hyperparathyroidism, and raised doubts about the justification for the clinical use of NIR imaging in parathyroidectomies [119]. Wolf et al. reported in a retrospective observational study that 57 of 66 (86.4%) histologically proven neoplastic glands from patients with primary hyperparathyroidism and 36 of 42 (85.7%) PTGs from patients with secondary hyperparathyroidism exhibited AF [121]. McWade et al., however, found significantly lower AF intensities in hyperfunctioning PTGs in primary hyperparathyroidism (pHPT) [98]. Kose et al. confirmed these findings and additionally noted a heterogeneous AF pattern in 75% of hyperfunctioning PTGs [122]. Several studies reported significantly reduced PTG detection rates, specifically in patients with secondary hyperparathyroidism (sHPT) compared to pHTP and thyroid diseases [99,119,121]. Furthermore, Squires et al. also found lower AF intensity in the PTGs of six patients with a pHPT affected by the rare genetic multiple endocrine neoplasia type 1 (MEN1) disorder compared to the PTGs of 65 patients with non-MEN1 sporadic pHPT [123]. The so-far-unknown AF molecule(s) might be differentially expressed in PTGs of different diseases.

Despite the technical feasibility and clinical usability of NIR AFI in (para-)thyroid surgery, further studies are needed to elaborate on the benefits for patients of routine application of AFI during (para-)thyroid surgery. Nevertheless, its use is today already widely regarded as an aid for PTG verification, rather than a stand-alone detection tool. Even though there are contradictory results regarding the benefit to patients by using NIR AFI, it can still be concluded that NIR AFI is at least a valuable asset for the training of “less experienced” surgeons.

### 2.6. Multiphoton Laser Tomography in Dermatology

Multiphoton laser tomography (MPT) is an SAF technique used for in vivo high-resolution examination of healthy and diseased skin tissue [124,125,126]. The commercially available multiphoton tomograph DermaInspect^®^ (JenLab GmbH, Berlin, Germany) includes a near infrared laser (excitation 750–850 nm), a galvo and piezo scanning system, and two photomultiplier tube detection modules. Unlike the so-far-described single photon excitation, MPT takes advantage of absorbing two near-infrared photons, emitting the same wavelengths, compared to one blue photon excitation (Figure 3). MPT detects fluorescence, e.g., from NAD(P)H and melanin in the epidermis, as well as elastic fibers and collagens in the dermis. MPT provides accurate measurements of the dermal matrix composition at different depths beneath the surface [127]. The DermaInspect^®^ has been used in multiple clinical settings, e.g., to characterize photoaging of facial skin and to provide information about the collagen content of the dermis layer. Koehler et al. found that MPT is a suitable technique for investigating qualitative and quantitative skin changes after ultraviolet B irradiation [128,129]. A three-month treatment with a cosmetic emulsion containing soy and jasmine enhances the signal level of the extracellular matrix (ECM), demonstrating the sensitivity of the MPT technique to non-invasively detect modifications in dermal collagen and elastin [130]. A similar device was introduced by Wang et al., who demonstrated in a clinical pilot study that pigmented skin lesions emit significantly brighter near-infrared fluorescence, whereas vitiligo lesions emitted no light upon excitation due to loss of skin melanin [131].

### 2.7. Fluorescence Lifetime Imaging Microscopy

Fluorescence lifetime imaging microscopy (FLIM) provides measurements on the lifetime of the autofluorescence, thus differing from standard autofluorescence measurements with the intensity signal alone. FLIM is based on the time for the fluorophore to decay from the excited electronic state to the ground state, called fluorescence lifetime (Figure 4). Lifetimes typically range between picoseconds (10–12 s) and nanoseconds (10–9 s), and are independent of the fluorophore concentration and excitation power. The data are computed and displayed as false-color images. This approach is helpful in providing quantitative fluorescence data in tissues in which fluorescence is scattered, e.g., in the skin (detailed information on the evolution of this technique and potential clinical application scenarios are reviewed elsewhere: [132,133]). Lifetime differences allow the discrimination of fluorophores assessed in the tissue as well as the identification of altered tissues in diseases. In dermatology, FLIM can be achieved for multiphoton tomography, e.g., to discriminate age of dermal tissue. Koehler et al. showed that young skin (mean age under 25 years) has a faster AF decay compared to old skin (mean age above 70 years). This was explained by the amount of collagen and elastin replaced by elastosis [125]. In gastroenterology, ex vivo lifetime analysis identified that neoplastic colonic polyps exhibit shorter lifetime values than normal tissue upon 435 nm excitation [134]. Recently, an in vivo clinical study applied FLIM endoscopy for oral lesions, indicating a contrast in cancerous vs. normal oral tissue, e.g., shorter lifetimes for collagen and NADH [135]. However, these encouraging results need to be validated by larger systemic studies.

### 2.8. Biomolecules Acting as Endogenous Fluorophores

A plethora of endogenous molecules with AF properties have been discovered. In general, AF molecules are considered to be either involved in metabolic functions or exert structural functions as intra- or extracellular tissue matrix proteins. Although experimental spectroscopic studies revealed the AF characteristics of single endogenous fluorophores, the initial discovery and exact contribution of particular fluorophores used for clinical purposes can be traced back only in few cases. When reviewing the current literature on clinical applications of AFI, we found that the origin of AF is mostly suggestive rather than supported by experimental evidence. This might be explained by the fact that several fluorophores with similar excitation properties are expressed in tissues of interest, and are therefore very difficult to distinguish. In addition, the clinically used AFI devices deploy such broad excitation and emission spectra (Table 1) that the resulting clinically visible phenotype reflects merged AF spectra for several fluorophores.

The excitation wavelengths range from ultraviolet light, typically used for excitation in endoscopic imaging, to the infrared wavelength ranges applied by the multiphoton technique introduced in dermatology. Some fluorophores, like the collagen adduct AGE, detectable in skin tissue, were identified decades ago [38]. The most frequently used medical device to detect SAF, the AGE Reader, still applies broad excitation (300–420 nm) and emission wavelength ranges (420–600 nm), so the resulting AF value is unlikely to reflect the quantity of a single fluorophore. Similarly, fluorophores and their respective ideal excitation wavelengths at other tissues also give rather vague estimates about AF sources. Taking the AF detected in the larynx, the study by Winiarski et al. found distinct AF emission spectra (in the ultraviolet and blue light wavelength range) between healthy or dysplastic tissues, which led the authors to create a comprehensive list of putative fluorophores (without assurance of correctness or completeness in the presented clinical cases) [85]. The following candidate fluorophores, excited at 290 or 370 nm wavelengths, were listed: Tyrosine (max. emission: 310 nm), tryptophan (max. emission: 340 nm), NAD(P)H (max. emission: 460 nm), collagen cross-links (max. emission: 420–460 nm), and flavin adenine dinucleotide (FAD) (max. emission: 520 nm). In cancerous lesions, the intensity of the AF signal decreases compared to in healthy tissue, mainly due to the reduction in flavins, breakdown of collagen cross-links, and blood absorption. Overall, this is an outstanding example of the complexity of AF detection in clinical scenarios, without calling the diagnostic benefits into question.

Similarly to SAF, pulmonary AF is typically excited in the UV wavelength range, and the emitted light is processed into a pseudo-color image. The main fluorophores in lung tissue are elastin, Tryptophan, NADH, and FAD, all of which decrease in cancerous tissue [136]. In addition, in acute cellular-rejection-related inflammation after lung transplantation, AF signals are supposed to originate in perivascular lymphocytes (without ultrastructural proof) [35]. The characteristic AF spectra in oral medicine include the 390 nm band, which is attributed to collagen, a second peak AF at 460 nm, which is associated with NADH emission, and, finally, a peak AF emission around 635 nm, believed to be due to endogenous porphyrins [137].

Although clinical AFI uses blue light excitation, experimental studies on AF signals in intestinal tissues found peak excitations at 488 and 561 nm, and the emitted light was detectable at 500–550 or 570–620 nm, respectively [2,7]. The intestine is covered by a layered wall with AF sources located in mucosal and submucosal layers. A potential mucosal fluorophore, besides cellular metabolites such as NAD(P)H, could be lipofuscin, which was found to be present in epithelial cells on the base of the crypts and also in tissue macrophages, which might explain the differential AF signals between inflamed and non-inflamed intestinal tissues (Figure 1) [138,139]. Regarding the submucosal compartment, ubiquitously expressed collagen is considered to be the main fluorophore [140]. In addition, neoplastic and regenerated tissues are characterized by an unphysiological abundance of extracellular matrix proteins, including collagen [141]. While the histological analysis can clearly differentiate between mucosal and submucosal AF, the impact of submucosal AF during endoscopy may be attenuated. Another suggested source of intestinal AF in inflammation and neoplasia could originate from altered blood flow and blood distribution in inflamed or neoplastic tissues [142]. The NIR fluorophore(s) found in PTGs remain unknown, and conclusive suggestions about the source are likely incorrect, as no blueprint exists for the NIR excitation range—unlike for the other clinical AFI application scenarios. Based on the current evidence, PTG AF originates in a molecule that is significantly less abundant in the surrounding thyroid tissue [95,143]. Furthermore, it is probably only a single molecule that is excited by NIR light, as the excitation maximum was found to be in a narrow wavelength range (814−826 nm) [95]. Calcium-sensing receptors and vitamin D receptors are the repeatedly hypothesized PTG fluorophore candidates, based on their expression in PTGs, surrounding tissues, and other organs [98,101]. However, scientific proof is still pending.

Precise knowledge of a specific fluorophore appears less relevant for clinical AFI in comparison with the eminent importance of the quantitative AF intensity, usability, and reliability of AFI techniques. However, in some cases, the expression of certain fluorophores could guide a clinical decision [144]. It still appears worth recapitulating the different fluorophores to foster the interdisciplinary evolution of AF methods in clinical use as well as for academic purposes, in order to study the physiological and pathophysiological mechanisms on the molecular level and to identify new specific, diagnostically usable properties in health and disease states. Table 2 summarizes the proposed endogenous AF fluorophores excited in clinical AFI, their excitation and emission spectra, and their diagnostic relevance.

The recorded AF signal within a human tissue represents the sum of light emitted by several fluorophores. Therefore, the tissue AF will depend on the concentration and quantum yield of the expressed fluorophores, on the presence of chromophores (principally hemoglobin) that absorb excitation and fluorescence light, and on the degree of light scattering that occurs within the tissue [145]. AF signals therefore reflect the biochemical and structural composition of the tissue, and are thus altered when tissue composition is changed by disease states, such as inflammation or neoplasia. The molecular changes, metabolic alterations, and grade of inflammation in neoplastic tissues are manifold and, therefore, the resulting alterations in tissue AF are multicausal.

## 3. Discussion

We compiled a systematic review on clinical applications that detect AF from endogenous fluorophores in patients as diagnostic tools or for medical guidance. AFI methods are well established in the diagnosis of degenerative eye diseases, like age-related macular degeneration. Current clinical practice guidelines and recommendations encourage inclusion of AFI in the diagnostic work-up to aid ophthalmologic diagnostics [162,163]. Although there is a growing body of evidence on the usability of AFI methods in clinical applications, the general acceptance is still quite low. The main reasons for that are the lack of systematic development, as well as validation strategies that prevent the evolution of standardized AFI techniques and the running of randomized clinical validation trials that allow comparison of study results even if different AFI techniques were applied. It is furthermore challenging to implement or even test novel approaches if the standard diagnostic methods are already highly accurate.

In some disciplines, reasonable progress has been made, and AFI is considered to be an improvement in existing diagnostic accuracy. In oral medicine, according to a German guideline, AFI can be performed during panendoscopy in patients with suspected lesions at clinical examination to identify possible synchronous secondary tumors [164]. An American guideline did not recommend AFI as a triage tool for the evaluation of potentially malignant disorders in primary care settings, due to low-quality evidence that resulted from the often-analyzed secondary care settings, where the prevalence of disease could modify the diagnostic test accuracy of these adjuncts [165]. A Cochrane review on diagnostic tests for oral cancer concluded that none of the available alternative methods, like vital staining, oral cytology, or light-based detection, can replace scalpel biopsy with histology; however, the non-invasive tests warrant further investigation as adjunctive tests [166]. Assessing the AF of the skin was found to correlate with blood glucose homeostasis and cardiovascular events. However, so far, SAF imaging is not implemented in national or international clinical guidelines for prediction of diabetes or risk assessment of diabetic complications. Harmonization of SAF results by leading scientists and physicians from the relevant national and international societies is desired and may be the first step towards a general implementation. Guidelines in thyroid surgery do not consider intraoperative AFI either, but rather encourage surgeons to dissect meticulously for PTG preservation [167]. In the future, given the results of this review, intraoperative AFI may also only play a supporting role in clinical practice in thyroid surgery. However, the potential benefits for training inexperienced surgeons or the ex vivo identification of PTGs in (accidentally) resected thyroid specimens appear as useful clinical application scenarios worth being further explored. A range of studies already explored the addition of AFI to endoscopic approaches for gastrointestinal diseases. However, the application of AFI, alone or as part of the ETMI procedure, is not mentioned in clinical guidelines for the diagnosis of GI tract malignancies [168,169,170]. Here, our review reflects AFI’s heterogeneous results in detecting gastrointestinal inflammation or neoplastic malignancies. While detection of mucosal inflammation by AFI is highly accurate, AFI performance is not superior to WLI examination. The benefits of AFI diagnostics in detecting intestinal inflammation were predominantly present in less experienced observers. This fact might be explained by the strong macroscopically visible phenotype of mucosal inflammation. In contrast, the discrimination of mucosal dysplasia from normal mucosa lacks accuracy to justify its routine application. Slightly similarly to thyroid surgery, AFI adds a benefit mainly for inexperienced endoscopists in finding the right diagnosis, while experienced endoscopists may not benefit clearly from advanced imaging techniques [26].

### 3.1. Future Directions

AFI is of increasing interest in various clinical disciplines, as it represents a fast, reliable, and cost-effective method. In the future, technical advancements will allow high-resolution AF detection, the specific excitation of fluorophores, and detection of emitted light within narrow wavelength ranges. In addition, experimental studies also already found that metabolic processes can be tracked by changing AF characteristics, caused, for example, by changing fluorescence spectra of NAD(P)H in its bound or free form or redox state [171]. A summary of the experimental advancements and first translational results on AF-based metabolic imaging was recently presented by Croce et al. [172]. Bridging the gap between experimental findings and their translation into clinical applications is challenging, especially for complex approaches that require expert knowledge on both sides—the technical (to operate devices) and the medical (to treat patients). Furthermore, basic science and experimental developments might not necessarily meet a clinical need [173]. An explicit limitation of AFI is the huge heterogeneity of AFI devices in terms of technical specifications as well as settings to acquire images, and the subsequent image data handling and analysis allow no comparability between studies, unless the same observer used the same device for AF image acquisition. 

Solutions for standardized imaging and software-based image analysis were presented by several workgroups. However, no obvious attempt to harmonize image acquisition and analysis has been made. Krauss et al. developed an algorithm that calculates a normalized AFI (NAFI) signal from ETMI data [174]. While NAFI could not differentiate accurately between inflamed and normal esophageal or gastric mucosa, tumorous mucosa was precisely detected. Automated image analysis software tools have been developed and tested. Quang et al. developed a two-class linear discriminant algorithm and tested its ability to discriminate between neoplastic and non-neoplastic regions of biopsied or surgical resected oral lesions [175]. The automated algorithm was trained to discriminate between healthy and affected tissues by the red-to-green ratio of the AF images. In addition, automated analysis of high-resolution microendoscopy images was tested. The algorithm correctly classified 100% of non-neoplastic and 61% of neoplastic sites in histologically examined suspect oral tissues, either from a punch biopsy or an excised surgical specimen [175]. These findings led the authors to conclude that automated image analysis may be especially useful in situations where an expert clinician is not available, and warrant further studies to test whether implementing the real-time automated image analysis may impact the clinician’s decision-making process. Attempts to use computational machine learning approaches for automated image analysis revealed a sensitivity and specificity of 59% (47%–70%) and 76% (68%–84%), respectively, in detecting caries lesions [176]. Software solutions for automated AF image analysis of colorectal lesions had also been tested by several workgroups, and were found to be able to discriminate non-neoplastic from neoplastic colorectal lesions with high sensitivity (94.2) and specificity (88.9) [177]. According to Inomata et al., the automated analysis of colorectal lesions could be helpful in distinguishing between neoplastic and non-neoplastic lesions and to additionally predict the depth of invasion by the color tone analysis in colorectal neoplasia [178]. However, the use of different AFI systems under very heterogenous imaging conditions and computational image modifications, like noise-reduction algorithms, higher frame rate, and more or less intense light sources, strongly limits the comparability between studies. Future studies should aim to harmonize the different attempts and to define valid diagnostic cut-off values, applicable independently of the AFI device.

A prototypic example of the evolution of an AFI device towards diagnostic usability is given by the AGE Reader (Diagnoptics Technologies B.V.). It enables measurements under standardized conditions by illuminating a defined skin area (guarded against surrounding light), including control readings and reference adjustments. It provides a quantitative SAF test result in a single number. Although the final measurement result is given in arbitrary units, a range of clinical validation studies provided ranges and cut-off values to grade the extent of skin AGE accumulation as a surrogate for diabetic or cardiovascular risk assessment. A further advantageous feature of the AGE Reader is its user-friendly handling and fast SAF acquisition—desirable characteristics for clinical applications. The AGE Reader is a small device that does not require much space, can be easily carried around, is affordable (one-time investment of approximately 7000.00 euros), and does not require expensive consumables. Finally, the AGE Reader is a great example of clinically desirable measurement modalities that do not require the excitation of well-defined individual compounds, but rather reflect the group reactivity of a range of fluorophores. Future developments are still required to overcome the present limitations, like the distortion of test results in darker-skinned patients due to lower ultraviolet reflectance or ethnicity-, age-, and gender-specific SAF characteristics [179,180]. The scope of application may also be extended. Pol et al. observed increased postoperative SAF values in patients undergoing colorectal surgery, which correlated with the duration of operation, intraoperative blood loss, and the postoperative C-reactive protein level [181].

In general, virtually no clinical study pinpoints the exact fluorophores addressed by the clinical AFI. Studies rather give rough estimates of the potentially excited fluorophores, referencing experimental studies. An exception represents the unknown fluorophore that illuminates PTGs upon excitation in the NIR wavelength spectrum. While endocrine glands (including PTGs) are well known for their AF characteristics upon excitation in the UV to visible green light wavelengths [182,183,184], only vague suggestions about the molecular NIR AF source of PTGs exist [101]. The benefits of the intraoperative AFI in detecting PTGs are controversial for various reasons, despite the fact that several studies reported that PTGs could be intraoperatively identified by NIR AFI before the surgeon saw them with the naked eye [107,108,111]. One limitation is the prolonged operation time, especially with AFI techniques that require switching the operation room light off. Benmiloud et al. quantified the duration of the operations, which was significantly longer in the NIR AFI group compared to the standard-care group (eight minutes between median values of both groups) [111]. Several attempts at intraoperative real-time monitoring of PTGs without delaying the operation were developed, and may even allow optimization of operation times without waiving intraoperative NIR AFI PTG detection [112,113,115,143]. A second limitation is that, foremost, only inexperienced surgeons profit from the intraoperative AFI guidance, requiring a case-to-case decision of whether or not to use intraoperative NIR AFI. Following Squires et al., a significant increase in the surgeon’s confidence in PTG identification when adding NIR AFI to the parathyroidectomy procedure was found [120]. It is worth thinking about differentiated recommendations for dedicated groups and for teaching purposes. It should further be considered that due to the multitude of available PTG AFI devices, the generalization of the study results is constrained. A standardized NIR AFI image acquisition and analysis is desirable and should be validated in in vivo settings and larger study cohorts. Once this is achieved, targeted clinical application of NIR AFI could be tailored for the patients’ benefit.

### 3.2. Further Experimental Trends with Translational Potential

In addition to the clinical AFI applications discussed so far, a range of promising experimental findings, with a strong potential to be translated into clinical applications, were published. The tracking of drugs with AF properties (like AEZS-108) could be useful for following their cellular localization and decay. AEZS-108 emits light upon excitation in the green wavelength spectrum, accumulates in circulating tumor cells, and its AF signal allows the dose-response analysis and might also be applicable to decipher off-target effects [185,186]. As presented before, AF techniques can be used in thyroid surgery, but might also be a promising method for margin status assessment in breast cancer surgery. AF and reflectance spectroscopy analysis of surgical specimens showed an excellent classification between normal or tumor tissues [187]. Despite the existing limitations of AFI, such as depth of tissue penetration, the authors conclude from their exploratory study that AFI and reflectance spectroscopy could be valuable for examining the superficial margin status of excised breast cancer specimens. AFI techniques show great further diagnostic potential-in addition to many other advanced endoscopic imaging techniques—in bladder cancer [188]. AF spectra assessment could also be used to evaluate intervertebral disc compression and degeneration [189]. Finally, multiphoton tomography (MPT) may be applicable to differentiate normal from abnormal spermatogenesis [190]. A steadily growing research field is the assessment of the metabolic states of cells by AFI, which is especially sensitive in reflecting the energetic and redox status of cells as well as tissues or organs, like the liver of murine animal models [172]. The translation for diagnostic purposes is currently limited due to the lack of appropriate techniques, which might, however, be available in the near future.

There is a need for further improvement of imaging tools, a consensus on unified AFI assessment procedures, and data analysis algorithms that strive to improve the accuracy of AF detection. Given the manifold promising experimental and clinical developments of AFI techniques to distinguish between healthy and diseased states during the last decade, AFI appears to be an easy-to-use and highly promising technique for increased diagnostic accuracy in a range of clinical disciplines. Its implementation as standard procedure is only a matter of time.

## 4. Materials and Methods 

A systematic search of the MEDLINE and Cochrane Library databases was conducted in accordance with the PRISMA guidelines [191]. As the particular focus of this review includes the recent advances in clinical applications of AFI (in extra-ophthalmic tissues), we considered only clinical studies published between January 2010 and December 2019. The search term “autofluorescence” was combined with terms relating to the clinical disciplines or medical conditions “reflux disease”, “inflammatory bowel disease”, “gastritis”, “esophageal cancer”, “gastric cancer”, “colorectal cancer”, “lung cancer”, “skin”, “diabetes”, “cardiovascular disease”, “caries”, “oral cancer”, “thyroid surgery”, “parathyroid surgery”, and “dermatology”. Following the database search, duplicates were excluded, and titles and abstracts were screened for eligibility. The remaining articles were reviewed in full text for confirmation of an AF technique as the main investigated method. Studies that were written in languages other than English, contained no human data or no original data, and case reports were excluded. In total, 1097 articles were identified, of which 113 articles were eligible to be included in this review (Figure 5).

## Figures and Tables

**Figure 1 molecules-25-02095-f001:**
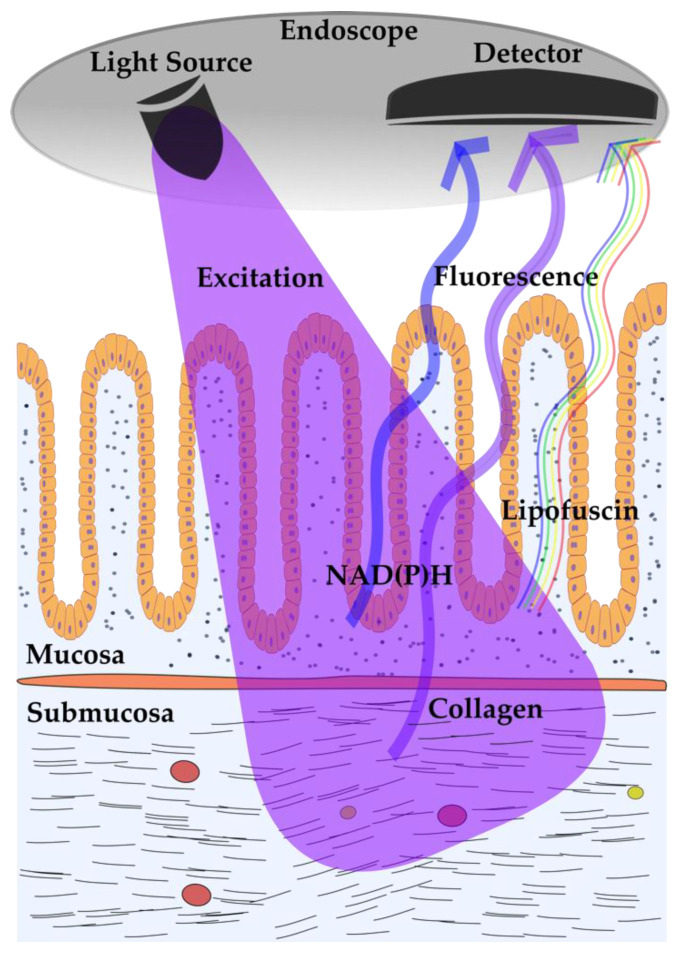
Schematic endoscopic autofluorescence imaging (AFI) of normal colonic tissue. In AFI mode, blue light excitation is used to excite local mucosal fluorophores, e.g., NAD(P)H or lipofuscin, and submucosal fluorophores like collagen. The endoscope is equipped with a light source and a detector to collect the emitted fluorescence light.

**Figure 2 molecules-25-02095-f002:**
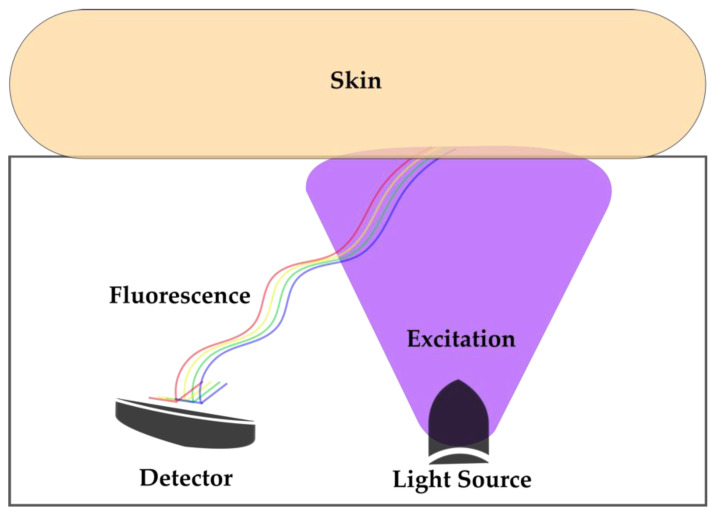
Schematic of skin autofluorescence. Upon blue light excitation of the skin, fluorescence of a wide spectrum is emitted by advanced glycation end products (AGEs) and detected by an autofluorescence (AF) device, e.g., the AGE Reader. The AGE fluorescence intensity positively correlates with the cardiovascular risk.

**Figure 3 molecules-25-02095-f003:**
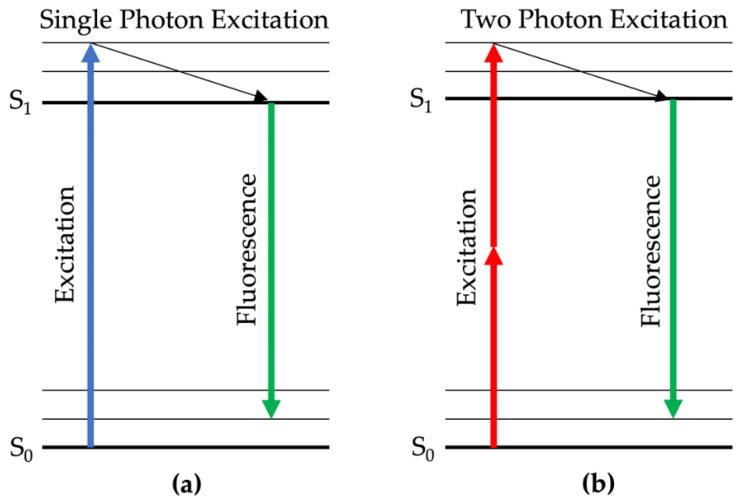
Jabłoński diagrams of single vs. two photon excitation principles: (**a**) In single photon excitation, the excitation wavelength is shorter than the fluorescence wavelengths; (**b**) in two photon excitation, the wavelengths of two exciting photons are longer then the resulting fluorescence. Multiphoton laser tomography (MPT) uses two near-infrared (NIR) photons to excite dermal fluorophores.

**Figure 4 molecules-25-02095-f004:**
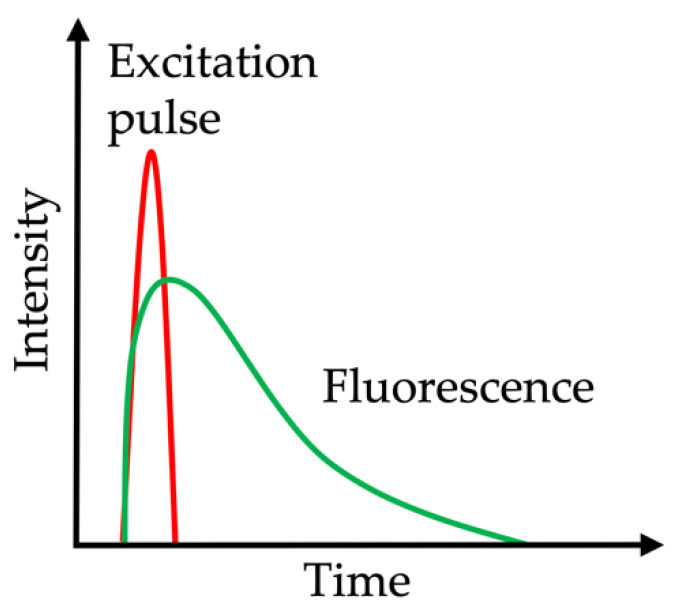
Schematic of the fluorescence lifetime imaging microscopy (FLIM) principle. A short excitation pulse (red) is followed by a longer fluorescence signal (green). FLIM measures the time of the fluorescence signal.

**Figure 5 molecules-25-02095-f005:**
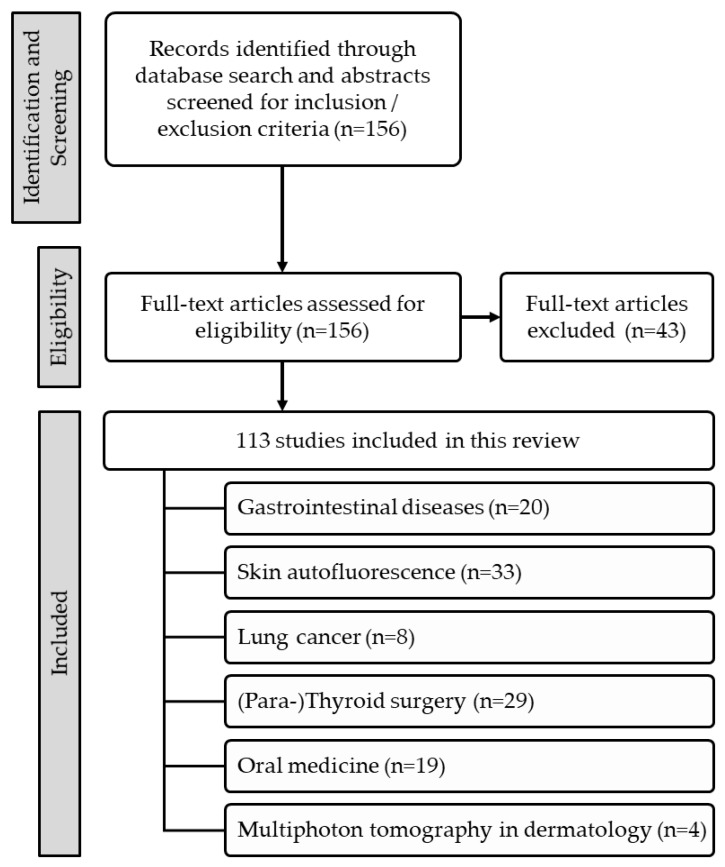
Flow diagram of study selection for this review.

**Table 1 molecules-25-02095-t001:** Overview of the excitation wavelength used for the different clinical applications.

Clinical Specialty	Excitation Wavelength Ranges	Clinical Imaging Procedure
Diseases with impaired redox homeostasis (e.g., diabetology, nephrology)	300−420 nm (ultraviolet light, peak at 370 nm)	Non-invasive skin tissue imaging
Oral medicine	400−460 nm (blue light)	Non-invasive oral imaging
Gastroenterology	390−470 nm (blue light)	Endoscopy
Pulmonology	488 nm (green light)	Endomicroscopy
Dentistry	655 or 780 nm (red or near-infrared light)	Caries screening
Thyroid surgery	690−770 nm (near-infrared light)	Intraoperative imaging
Dermatology	750−850 nm (multiphoton principle)	Multiphoton imaging

**Table 2 molecules-25-02095-t002:** Overview and characteristics of commonly known endogenous fluorophores in clinical usage, their typical AF properties, and diagnostic values. All wavelengths are approximate values of the optimal excitation and emission spectra.

Fluorophore	Function	Excitation (nm)	Emission (nm)	Diagnostic Value, Esp. Changes Depending on Progress of Disease	References
Collagen	Structural protein in ECM	330–340	400–410	Altered in neoplastic lesions or fibrotic states, e.g., in liver fibrosis	[146,147,148,149,150]
Elastin	Structural protein in ECM	350–420	420–510	Altered expression in invasive tumors	[151]
Keratin	Structural protein in ECM	355–405	420–480	Altered expression in invasive tumors, delineate tumor borders	[152,153]
NAD(P)H	Cofactor in redox reactions	330–380	440 (bound); 462 (unbound)	Biomarkers of energy metabolism and redox state	[149,150,154,155]
FAD	Cofactor in redox reactions	440–450	525	Biomarker of energy metabolism and redox state e.g., in cancer formation	[85,156,157]
Porphyrin	Formation of heme	405	630–700	Altered in dental caries and neoplastic lesions	[158,159]
Lipofuscin	End product of lysosomal digestion	400–500	480–700	Biomarker of degenerative diseases	[160]
AGEs	Metabolic by-products	300–420	420–600	Accumulate with age and progressive degenerative diseases	[38,39,161]
Amino acids, e.g., Tyrosin, Tryptophan	Protein metabolism	<310	>500	Altered abundance in invasive tumors	[81,150]
Calcium-sensing receptor	Regulation of parathyroid hormone (PTH) secretion	785	822	Postulated candidate fluorophore for parathyroid AF, distinguishing between parathyroid and surrounding tissues	[98]

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
