# Peer review of "Recent Advances and the Potential for Clinical Use of Autofluorescence Detection of Extra-Ophthalmic Tissues"

_molecules, 2020, doi:10.3390/molecules25092095_

Round 1

Reviewer 1 Report

Manuscript Number: molecules-774342

Ms. Title: Recent advances and the potential for clinical use of autofluorescencedetection of extra-ophthalmic tissues

This review summarizes the clinical applications of autofluorescence for 2010-2019. Both the advantages and disadvantages of autofluorescence detection were also well explained. Sensing with fluorescence from endogenous molecules (autofluorescence) has become important in medicine, and the contents of this review will be of interest to readers of Molecules. I would like to raise the following three points for the authors to be revised before publication.

(1) Number of measurements (n) and p-value (p) are italic (e.g., page 5) and the text should be Roman type in section 2.6, page 13.

(2) In page 14, the authors include NAD+ as representative candidate fluorophores. This is mistake because NAD+ is known to show no fluorescence (e.g., M. Y. Berezin and S. Achilefu, Chem. Rev. 110 (2010) 2641). The authors should check and delete NAD+ in page 14.

(3) In this review, the authors focused on the detection of the "intensity" of autofluorescence. However, the "lifetime" of autofluorescence is also widely used in medicine. For example, two reviews showing application of autofluorescence lifetime in medicine were published (S. Coda et al., Endosc. Int. Open. 3 (2015) E380 and E. A. Gibson et al., J. Ophthalmol. (2011) 870879). Fluorescence lifetime is a more quantitative parameter than fluorescence intensity because the value of fluorescence lifetime does not depend on photobleaching or excitation intensity. Lifetime techniques are particularly useful for skin because light scattering on the skin surface prevents quantification of fluorescence intensity. I recommend that the authors create one sub-section and mention the application of autofluorescence lifetime to medicine.

Author Response

Reviewer 1 (Round 1):

This review summarizes the clinical applications of autofluorescence for 2010-2019. Both the advantages and disadvantages of autofluorescence detection were also well explained. Sensing with fluorescence from endogenous molecules (autofluorescence) has become important in medicine, and the contents of this review will be of interest to readers of Molecules. I would like to raise the following three points for the authors to be revised before publication.

(1) Number of measurements (n) and p-value (p) are italic (e.g., page 5) and the text should be Roman type in section 2.6, page 13.

Answer: We have formatted all measurements (n) and p-values (p) italic and Section 2.6, page 13 is now written in Roman type. All changes have been marked in red in the revised version of the manuscript.

(2) In page 14, the authors include NAD+ as representative candidate fluorophores. This is mistake because NAD+ is known to show no fluorescence (e.g., M. Y. Berezin and S. Achilefu, Chem. Rev. 110 (2010) 2641). The authors should check and delete NAD+ in page 14.

Answer: We agree, that NAD+ shows no fluorescence. Therefore, NAD+ has been deleted from the respective section in page 14.

(3) In this review, the authors focused on the detection of the "intensity" of autofluorescence. However, the "lifetime" of autofluorescence is also widely used in medicine. For example, two reviews showing application of autofluorescence lifetime in medicine were published (S. Coda et al., Endosc. Int. Open. 3 (2015) E380 and E. A. Gibson et al., J. Ophthalmol. (2011) 870879). Fluorescence lifetime is a more quantitative parameter than fluorescence intensity because the value of fluorescence lifetime does not depend on photobleaching or excitation intensity. Lifetime techniques are particularly useful for skin because light scattering on the skin surface prevents quantification of fluorescence intensity. I recommend that the authors create one sub-section and mention the application of autofluorescence lifetime to medicine.

Answer: We agree, that lifetimes are clinically used and would add a value to our manuscript. Therefore, we included a new section (2.7) regarding basics of fluorescence lifetimes and its clinical use in the revised version of the manuscript. In addition, a figure illustrating the FLIM priniciple has been added.

Thank you for reviewing our manuscript.

Reviewer 2 Report

Paper summary: The manuscript summarizes autofluorescence techniques in clinical imaging across several disease types outside of ophthalmology. An extensive search was conducted to identify studies, with 113 studies ultimately included in the manuscript. The summary of findings in this manuscript will be useful for collating a general clinical consideration for autofluorescence imaging for specific disease types.

Line-specific comments:

  1. Line 633: Possible error – “The following candidate fluorophores, excited at 290 or 370 nm wavelengths, were listed: tyrosine (max. emission: 310 nm), tryptophan (max. emission: 340 nm), NAD+ (max. emission: 460 nm), NAD(P)H (max. emission: 460 nm), collagen cross-links (max. emission: 420 - 460 nm) and FAD (max. emission: 520 nm).” It was this reviewer’s understanding that NAD+ is nonfluorescent, while NADH/NAD(P)H are fluorescent.
  2. Line 769: Please add examples – “Automated image analysis software tools have been developed and tested.”
  3. Line 777: What is considered “good”? – “Attempts to use computational machine learning approaches for automated image analysis revealed a good discriminatory ability to detect caries lesions [167].”

General comments:

  1. There were some examples of "flowery" or imprecise language (e.g. Line 696 - "We herewith...or Line 562 - "AF seems to be..."). It would be beneficial to give the paper a once over to revise a bit more.
  2. It would be helpful to occasionally write out frequently used acronyms for easier reading. I realize there is a list of abbreviations at the beginning of the manuscript, but writing out the full abbreviation occasionally could help the paper flow better without having to frequently fall back to the list.
  3. I'm not sure whether this was intentional, but the entire paragraph beginning at Line 587 is bolded and italicized. 
  4. In Table 2, there is no mention of FAD, a dominant autofluorescent contributor in redox imaging of cancer.
  5. Figure 3 is difficult to parse – it would be beneficial to the reader to better demarcate the different clinical stages of implementations and differing areas of implementation (i.e. parathyroid, skin, GI, etc.). How is “evident clinical benefit” defined? Is it just increasing levels of specificity of detection compared to traditional clinical approaches? Or is it improved patient outcomes? Can references for each AFI scenario be included on the Figure?
  6. In the Methods, the primary search term was “autofluorescence”. Were other keywords used? For instance, “label-free fluorescence” is often used in place of autofluorescence, as is “endogenous fluorescence.”

Author Response

Paper summary: The manuscript summarizes autofluorescence techniques in clinical imaging across several disease types outside of ophthalmology. An extensive search was conducted to identify studies, with 113 studies ultimately included in the manuscript. The summary of findings in this manuscript will be useful for collating a general clinical consideration for autofluorescence imaging for specific disease types.

Line-specific comments:

1. Line 633: Possible error – “The following candidate fluorophores, excited at 290 or 370 nm wavelengths, were listed: tyrosine (max. emission: 310 nm), tryptophan (max. emission: 340 nm), NAD+ (max. emission: 460 nm), NAD(P)H (max. emission: 460 nm), collagen cross-links (max. emission: 420 - 460 nm) and FAD (max. emission: 520 nm).” It was this reviewer’s understanding that NAD+ is nonfluorescent, while NADH/NAD(P)H are fluorescent.

Answer: We agree, that NAD+ shows no fluorescence. Therefore, NAD+ has been deleted from section 2.7.

2. Line 769: Please add examples – “Automated image analysis software tools have been developed and tested.”

Answer: We added more detailed examples for “automated image analysis”, the paragraph reads now as follows: Automated image analysis software tools have been developed and tested. Quang et al. developed a two-class linear discriminant algorithm and tested its ability to discriminate between neoplastic and non-neoplastic regions of biopsied or surgical resected oral lesions [175]. The automated algorithm was trained to discriminate between healthy and affected tissues by the red to green ration of the AF images. In addition, automated analysis of high resolution microendoscopy images was tested. The algorithm correctly classified 100% of non-neoplastic and 61% of neoplastic sites in histologically examined suspect oral tissues, either from a punch biopsy or an excised surgical specimen [175]. These findings led the authors to conclude that automated image analysis may be especially useful in situations where an expert clinician is not available and warrant further studies to test whether implementing the real-time automated image analysis may impact the clinician’s decision-making process. Attempts to use computational machine learning approaches for automated image analysis revealed a sensitivity and specificity of 59% (47% - 70%) and 76% (68% - 84%) respectively to detect caries lesions [176]. Software solutions for automated AF image analysis of colorectal lesions had also been tested by several workgroups and found to be able to discriminate non-neoplastic from neoplastic colorectal lesions with high sensitivity (94.2) and specificity (88.9) [177]. According to Inomata et al., the automated analysis of colorectal lesions could be helpful in distinguishing between neoplastic and non-neoplastic lesions and to additionally predict the depth of invasion by the color tone analysis in colorectal neoplasia [178].

3. Line 777: What is considered “good”? – “Attempts to use computational machine learning approaches for automated image analysis revealed a good discriminatory ability to detect caries lesions [167].”

Answer: We now provide, the original published performance metrics, see page 22, line 793: “Attempts to use computational machine learning approaches for automated image analysis revealed a sensitivity and specificity of 59% (47% - 70%) and 76% (68% - 84%) respectively to detect caries lesions [176].”

General comments:

1. There were some examples of "flowery" or imprecise language (e.g. Line 696 - "We herewith...or Line 562 - "AF seems to be..."). It would be beneficial to give the paper a once over to revise a bit more.

Answer: We deleted “flowery” fillings and refined the wording, providing plain and precise language throughout the manuscript.

2. It would be helpful to occasionally write out frequently used acronyms for easier reading. I realize there is a list of abbreviations at the beginning of the manuscript, but writing out the full abbreviation occasionally could help the paper flow better without having to frequently fall back to the list.

Answer: We aimed to address this point during the revision but found that repeatedly introducing similar abbreviations leads to redundancy and creates more confusion than clarity. We double-checked with the Journal’s “Instruction for authors” which state: “Abbreviations should be defined in parentheses the first time they appear in the abstract, main text, and in figure or table captions and used consistently thereafter.” Therefore, we belief that sticking to the Journal’s instructions would appropriate, however if the Editor of the manuscript would also prefer to occasionally write out frequently used acronyms, we would include these modifications.

3. I'm not sure whether this was intentional, but the entire paragraph beginning at Line 587 is bolded and italicized. 

Answer: We have also identified and corrected this typesetting mistake in the revised version of the manuscript.

4. In Table 2, there is no mention of FAD, a dominant autofluorescent contributor in redox imaging of cancer.

Answer: We agree, FAD is a relevant fluorescing redox-active coenzyme. Therefore, we modified Table 2 by adding data for FAD.

5. Figure 3 is difficult to parse – it would be beneficial to the reader to better demarcate the different clinical stages of implementations and differing areas of implementation (i.e. parathyroid, skin, GI, etc.). How is “evident clinical benefit” defined? Is it just increasing levels of specificity of detection compared to traditional clinical approaches? Or is it improved patient outcomes? Can references for each AFI scenario be included on the Figure?

Answer: The figure was intended to give an easy-to-read overview of the main outcomes of the review and the progress towards clinical implementations of AF methods. We belief that „evident clinical benefit“ would lead to visible activities by medical societies to mention and include AF methods in clinical guidelines. The figure was the conclusion drawn by the authors of the review. We deleted the figure in the revised version of the manuscript.

6. In the Methods, the primary search term was “autofluorescence”. Were other keywords used? For instance, “label-free fluorescence” is often used in place of autofluorescence, as is “endogenous fluorescence.”

Answer: We agree, other terms like “endogenous fluorescence” also exist, however preferred in basic science articles. We finally focused on the primary term “autofluorescence”, which is the most established term, especially from the clinician´s view and that gives the most and accurately results for our manuscript. We know that the search results with your suggested search terms are partially overlapping, although we cannot exclude that there may be one or more studies, that would also fit in our manuscript. To condensate the workload for our wide-spread review we did a concise search, nevertheless believing strongly to demonstrate a profound analysis.  

Thank you for reviewing our manuscript.

Reviewer 3 Report

The manuscript, Recent advances and the potential for clinical use of autofluorescence detection of extra-ophthalmic tissues, focuses on the methodology of fluorescence specifically focusing on cellular information provided by autofluorescent molecules within cells/tissues. This area of research has garnered much attention over the past few years as it provides scientists with a method of examining the health and metabolic properties of cells without having to cause destruction to the system. As expected, such an application to cell/tissue characterization has opened the doors to the exciting possibly of in vivo use as a determine factor for metabolic disorders, such as diabetes. The authors do an excellent job in describing the application and readouts from AF in multiple scientific research fields and have no issues with the manuscript being published.

Minor Suggestion: The authors present a lot of data throughout the manuscript and do a great job in presenting the quantitative readouts. However, these sections could benefit from having a few figures. I know that getting permission to reprint figures might hinder this but I would recommend to the authors to draw a schematic diagram of the data collection/readout for a few of the sections listed.

Author Response

Reviewer 3 (Round 1):

The manuscript, Recent advances and the potential for clinical use of autofluorescence detection of extra-ophthalmic tissues, focuses on the methodology of fluorescence specifically focusing on cellular information provided by autofluorescent molecules within cells/tissues. This area of research has garnered much attention over the past few years as it provides scientists with a method of examining the health and metabolic properties of cells without having to cause destruction to the system. As expected, such an application to cell/tissue characterization has opened the doors to the exciting possibly of in vivo use as a determine factor for metabolic disorders, such as diabetes. The authors do an excellent job in describing the application and readouts from AF in multiple scientific research fields and have no issues with the manuscript being published.

Minor Suggestion: The authors present a lot of data throughout the manuscript and do a great job in presenting the quantitative readouts. However, these sections could benefit from having a few figures. I know that getting permission to reprint figures might hinder this but I would recommend to the authors to draw a schematic diagram of the data collection/readout for a few of the sections listed.

Answer: We agree and included additional figures in the revised version of the manuscript, visualizing data collection/readout.

Thank you for reviewing our manuscript.